# Single-vat single-cure grayscale digital light processing 3D printing of materials with large property difference and high stretchability

Liang Yue [1], S. Macrae Montgomery [1], Xiaohao Sun[1], Luxia Yu[1], Yuyang Song[2], Tsuyoshi Nomura[3], Masato Tanaka[2] & H. Jerry Qi [1] ✉

Multimaterial additive manufacturing has important applications in various emerging fields. However, it is very challenging due to material and printing technology limitations. Here, we present a resin design strategy that can be used for single-vat single-cure grayscale digital light processing (g-DLP) 3D printing where light intensity can locally control the conversion of monomers to form from a highly stretchable soft organogel to a stiff thermoset within in a single layer of printing. The high modulus contrast and high stretchability can be realized simultaneously in a monolithic structure at a high printing speed (z-direction height 1 mm/min). We further demonstrate that the capability can enable previously unachievable or hard-to-achieve 3D printed structures for biomimetic designs, inflatable soft robots and actuators, and soft stretchable electronics. This resin design strategy thus provides a material solution in multimaterial additive manufacture for a variety of emerging applications.

Additive manufacturing (AM), or 3D printing, allows the fabrication of structures with geometric and material complexities far beyond what can be created using traditional manufacturing techniques. New 3D printing capabilities have started to be used for functional applications in deployable structures[1], soft robotics[2], flexible electrical components[3,4], and biomimetic designs[5,6]. Many of these applications require the use of materials with vastly different properties, such as nature-like structures[7], airless tires ([https://michelinmedia.com/michelin-uptis/](https://michelinmedia.com/michelin-uptis/))[8], multistable absorbers[9,10], and 4D printing[8–10]. But fabricating highly stretchable soft materials and stiff materials in one AM process with high efficiency is very challenging.

Inkjet printing (IJP) is one of the most popular methods to achieve multimaterials in a single printed part[11,12] and have been extensively used in the past, although the cost for IJP printers with multimaterial capabilities is high. In addition, most of the IJP printable materials fail at low to moderate strain[13], making them unsuitable for applications

that require high stretchability. There have also been studies that mix multiple materials prior to extrusion[5,14,15] or employ multiple nozzles[16–18] in direct ink write (DIW) printing, but DIW printing is relatively slow and has limited accuracy compared to other AM techniques. When multiple materials cannot be combined using one AM process, the desired structures are often achieved by using multiple fabrication methods[2,19,20]. For example, DIW printing was used to write stiff fibers on prefabricated (by molding or 3D printing) soft elastomer structures[16,21]. Previously reported programmable curvatures of tube-like actuators with multimaterials were fabricated similarly by using multi-step processing with different techniques[16,19,22]. Overall, the multi-ink or multi-method AM techniques are often not efficient and have limited capability to fabricate structures with multimaterial properties with complicated distributions.

Digital light processing (DLP) is a high-speed and high-resolution printing method and has become increasingly popular in recent years.

[1]The George W. Woodruff School of Mechanical Engineering, Georgia Institute of Technology, Atlanta, GA 30332, USA. [2]Toyota Research Institute of North America, Toyota Motor North America, Ann Arbor, MI 48105, USA. [3]Toyota Central R&D Laboratories, Inc., Bunkyo-ku, Tokyo 112-0004, Japan. ✉ e-mail: qih@me.gatech.edu

It uses a projector to irradiate a thin layer of resin with images of the cross-sections of a solid part. In a typical DLP printing, a single resin vat is used, and only z-direction motion is needed to move the build plate. Photopolymerization (or photocuring) can occur in a few seconds. These make DLP one of the fastest 3D printing techniques. Nonetheless, the use of a single resin vat makes DLP, in general, not suitable for printing parts with multiple-material properties. Methods using multiple vats have been developed to print two or more materials by transferring a part between multiple vats[23,24]. Great care must be taken to avoid cross-contamination. Switching resin vats and cleaning significantly slow down the printing speed. In addition, the number of available material properties in multi-vat approaches is limited by the number of vats, making it hard to obtain a continuous transition of material properties. Boydston group developed a single-vat multiple-material printing method by using two different initiators and two different light sources[5]. However, the quality of printed parts relies on the alignment of two light sources and the tunable property range is limited in a single vat.

Recently, grayscale digital light processing (g-DLP) has emerged as a promising strategy to obtain a wide variety of material properties[25,26] and color[27] from a single resin material. In g-DLP printing, the local degree of monomer conversion is controlled by light intensity, which can be readily manipulated at pixel level by the input grayscale image. To obtain a large range of material properties, a special ink that contains bisphenol A ethoxylate diacrylate(BPADA), glycidyl methacrylate(GMA), and Jeffamine was developed in combination with a two-stage cure (or dual cure) strategy[25]. The first photocuring with controlled grayscale during 3D printing fixes the shape with a slight difference in the mechanical properties (modulus difference of 10–20 times). The second stage, thermally curing the printed part, boosts the differences in the material properties. By this method, Young's modulus, which spans nearly 1000 times, was achieved. It requires a long second-stage thermal curing. In addition, the final printed material has low stretchability (less than 35% in the soft state), thus limiting their functional applications. Besides, single-cured g-DLP was also used to modulate the printed properties[26,28,29]. It generally relies on varying the monomer conversion with different light to achieve the properties difference. However, it still remains a significant challenge to achieve high property difference and high stretchability simultaneously in efficient single-step printing.

In this paper, we present a design strategy of resins for single-stage g-DLP printing that possesses a very wide range of modulus as well as a very high stretchability. We use three different monomers in the resin where a stiff monomer helps to increase the stiffness at a high degree of cure (DoC) and two soft monomers with abundant side groups for hydrogen bonds facilitate high stretchability at the low DoC. The DoC can be readily manipulated by the light intensity in the g-DLP printing. The single-vat single-cure printed polymer can have a modulus ranging from 0.016 to 478 MPa (nearly 30,000 times difference) with a stretchability of up to 1500% at the soft state. Besides, the printing can be conducted at high printing speed (3 s per 0.05 mm layer, giving a build rate of 1 mm/min), and multiple parts can be printed simultaneously. To our best knowledge, this capability of rapid printing with large property differences and high stretchability has never been achieved before. The wide range of properties outperforms most inks used in inkjet 3D printing, multi-vat, as well as single-vat DLP printing. The soft state stretchability is among the best performance in single-material single-property printing. We further demonstrate such superiorities for the fabrication of various monolithic structures from anisotropic composites, biomimetic designs, inflatable structures, and customized wearable sensors. This capability for rapid prototyping of parts with dramatic property difference and superior stretchability thus has the potential for applications in pre-surgical models, soft robots, wearable electronics, etc.

## Results

### Multi-properties of single vat g-DLP printing

The photocurable resin ink is rationally designed. As illustrated in Fig. 1a, it consists of the monofunctional isobornyl acrylate (IBOA) and 2-hydroxyethyl acrylate (2-HEA) as the linear chain builders as well as the aliphatic urethane diacrylate (AUD) as the crosslinker and the reactive diluent (Fig. 1b–d and Supplementary Information (SI) Fig. 1 for chemical structures). The formulated ink has a low viscosity of around 0.05 Pa•s (Supplementary Fig. 2). AUD is a viscous oligomer with high molecular weight aliphatic chains and urethane units. It forms H-N•••O hydrogen bonds when interacting with 2-HEA and IBOA monomers, meanwhile, 2-HEA has abundant -OH groups that form additional O-H•••O hydrogen bonds as shown in Supplementary Figs. 3, 4. At low DoC, the covalent network with the prevalent hydrogen bonds provides high stretchability at the soft state, as shown in Fig. 1b. At high DoC, the stiff IBOA brings up the glass transition temperature ($T_g$) to be well above room temperature, yielding the glassy behaviors with the high modulus (Fig. 1c). Furthermore, the DoC can be conveniently controlled by light intensity, thus enabling the g-DLP printing to achieve drastic mechanical property differences within a single layer of printing.

We use a bottom-up DLP printer where light is projected from the bottom of the vat (Fig. 1e). As a demonstration, the design of a snail is sliced and processed into 2D grayscale images (Fig. 1e) using a MATLAB script (MathWorks, Natick, MA, USA), which controls the local property using the grayscale (or the ultraviolet (UV) light intensity) of each pixel. The grayscale images are then projected onto the ink-vat window from the bottom, initiating free radical polymerization. The oxygen-permeable Teflon (PTFE) window enables the easy separation of the cured polymer. This bottom-up continuous liquid interface production (CLIP)[30] approach allows for a rapid printing speed of 1 mm/min (3 s per 0.05 mm layer). The locally modulated UV intensity results in different DoCs, and thereby different mechanical properties, throughout the printed part. As shown in Fig. 1f, the printed snail possesses an integrated stiff shell (using 100% light intensity) and a soft body (using 40% light intensity). The shell holds 1 kg weight without visible deformation, while the body can be easily stretched by 400%.

The different material properties are controlled by the grayscale level of the UV light, which varies from 0% (full intensity, labeled as G0) to 100% (full dark, labeled as G100). The correspondence between grayscale and light intensity is presented in Supplementary Fig. 5. The photopolymerization kinetics is studied with a photopolymerization (PP) model[31] to analyze the correlation between the depth-dependent DoC at light does (SI, Section 1). Based on our slicing thickness of 50 μm and using the theoretically predicted correlation (Supplementary Fig. 6) and experiments, we use a grayscale range from G0 level to G70 (70% darkness) to ensure a rapid printing speed and good shape fidelity. At G0, the light intensity is 24.4 mW/cm$^2$ and the DoC is 94% (determined by FTIR measurement; Supplementary Fig. 7); at G70, the light intensity is 1.4 mW/cm$^2$, and the resulting DoC is 55%.

The mechanical and thermomechanical properties of printed samples with different grayscale levels are evaluated by uniaxial tensile tests and dynamic mechanical analysis (DMA; see Materials and Methods section for details). As shown in Fig. 2a, the printed polymer gradually becomes softer from G0 down to G50, with Young's modulus of 478 MPa at G0. The stiff state shows superior toughness, which is around 109 J/m$^3$. The fracture toughness is also measured by the tearing test and ranges from 650 to 10,000 J/m$^2$ (Supplementary Fig. 8). The polymer in the rubbery state using G60 and G70 possesses a modulus of 0.12 and 0.016 MPa, respectively, and can be stretched by near 1500% (Fig. 2b). The extensive presence of hydrogen bonding with crosslinked network makes the printed part in a stable organogel state with a gel fraction around 40 and 25% for G60 and G70 (Supplementary Figs. 9, 10). The soft polymer also exhibits excellent elastic properties and resilience even after 10,000 cycles of cyclic stretching

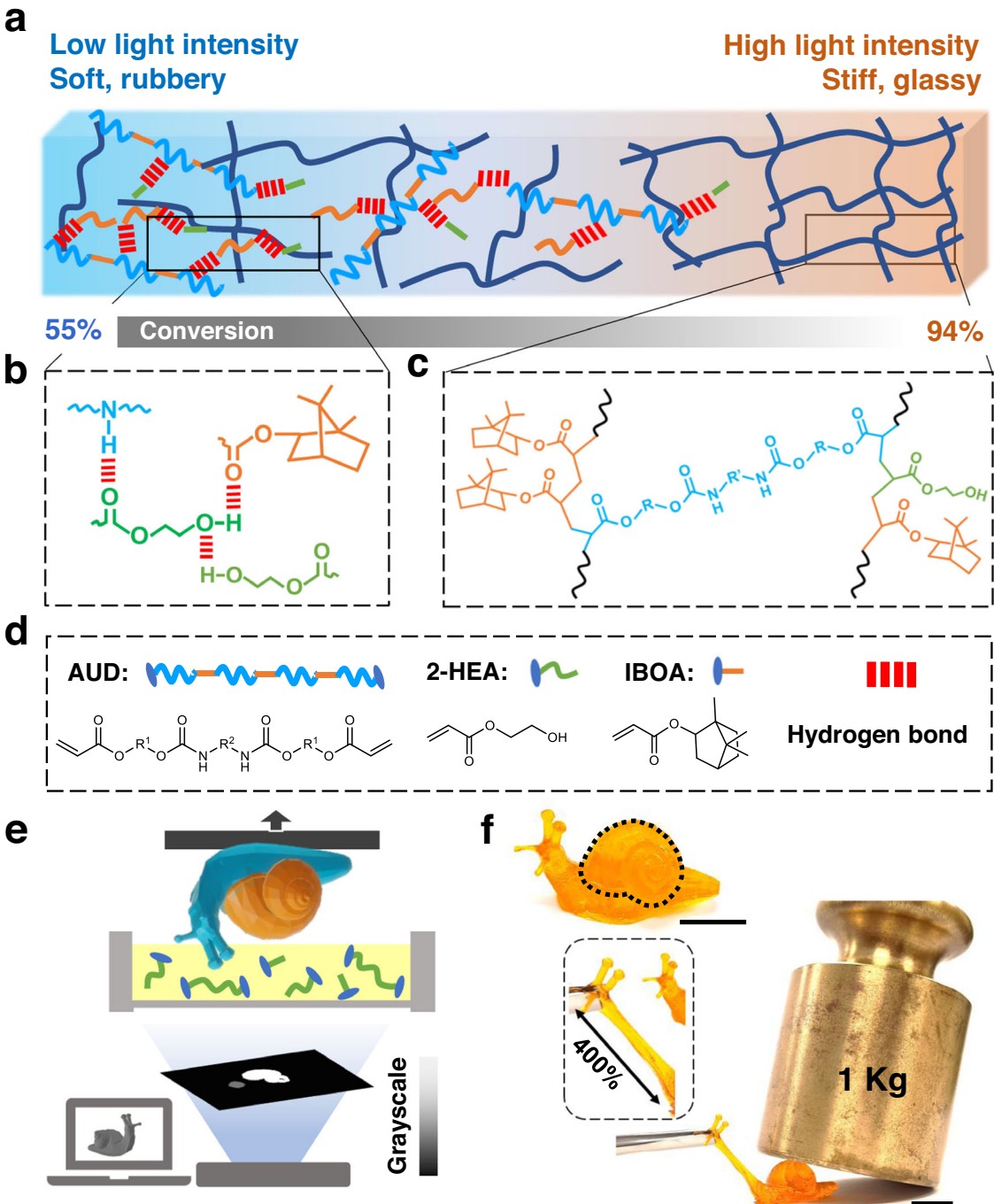

**Fig. 1 | Using light to control mechanical properties in g-DLP printing.**
**a** Schematic illustration of the crosslink density and properties with different
conversions. **b**, **c** Schematic illustrations of the hydrogen bonding and crosslinking
of the polymer network. **d** Chemical structure of the monomers; **e** Schematic of
bottom-up g-DLP 3D printing; **f** A g-DLP printed snail with a hard shell that can
support 1 kg weight and soft neck that can be stretched by 400%. The scale
bar is 1 cm.

at a high strain between 200 and 300% (Fig. 2c, d). To validate the
contribution of 2-HEA monomer to the elastic stretchability, we print a
reference sample without 2-HEA with the same DoC. The reference
sample (with only AUD and IBOA) exhibits similar rubbery behavior
but much lower elongation at break (~500%, Supplementary Fig. 11).
After the fatigue test with 10,000 cycles in the same strain range, the
residual strain is observed (Supplementary Fig. 11). There is an appar-
ent stress decrease due to network relaxation (Supplementary Fig. 12),
and the network is unable to fully recover in the absence of 2-HEA
monomer. To further validate the function of hydrogen bonding at low

DoC, we test a resin by replacing AUD and 2-HEA with PEGDA (poly(-
ethylene glycol) diacrylate, Mw ~700) and BA (butyl acrylate). In this
material, low DoC results in fewer crosslinking points and shorter lin-
ear chains. Without hydrogen bonding between the uncured mono-
mers and the crosslinked network, the samples are weak with low
stiffness and stretchability (Supplementary Fig. 13). We also observe
the same results in previously reported g-DLP resin that low DoC leads
to weak mechanical properties as expected[25].

The stiff monomer IBOA brings up the $T_g$ at high DoC and makes
the network stiff, which ensures the high modulus contrast at different

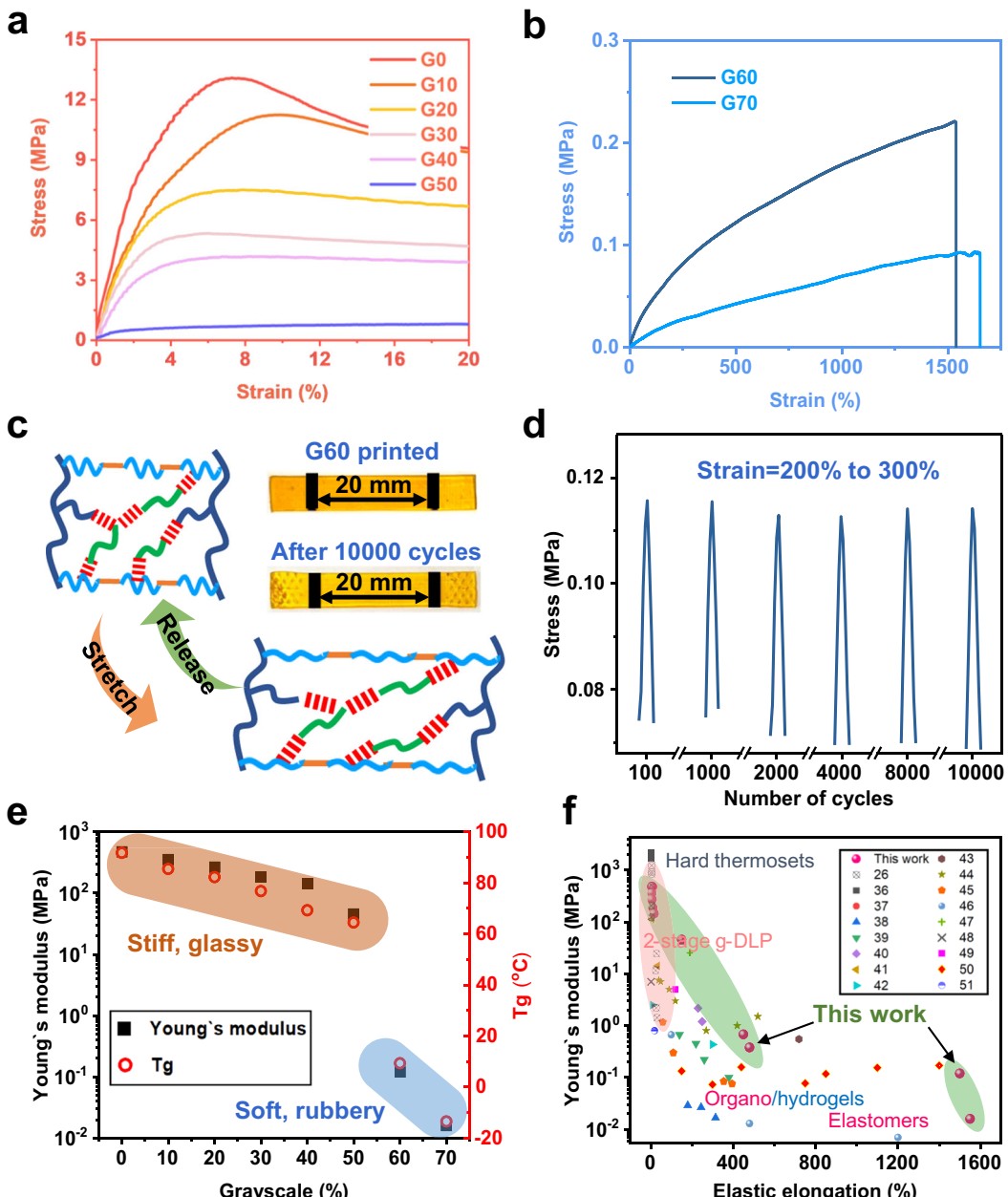

**Fig. 2 | Characterization of the printed material. a** Stress-strain curves of G50 to G0; **b** Stress-strain curves of G60 and G70; **c** Schematic of recovering of the network and photo comparison of G60 sample before and after 10000 fatigue cycles; **d** Fatigue test of G60 sample with an applied strain between 200 to 300%; **e** Correlation of Young's modulus and glass transition temperature with grayscale levels; **f** A comparison of the mechanical property range between the single vat g-DLP printing with general reported or commercially available DLP ink[20,25,35–49].

DoC. Figure 2e summarizes Young's moduli and $T_g$s from DMA (Supplementary Figs. 14, 15) at different grayscales, showing Young's modulus contrast between stiff G0 and soft G70 nearly 30,000 times. Figure 2f shows a comparison of our resin with reported DLP materials in literature, including two-stage cured hard thermoset, soft elastomers, hydrogels, and organogels[20,25,32–46]. As can be seen, our single resin covers the widest range of modulus and elastic elongation. It was also notable that the transition between the stiff thermoset and rubbery gel state was not continuous in this plot. This was because the printing properties were too sensitive to control at this threshold DoC range.

In our previous two-stage g-DLP printing, the modulus difference was less than 20 times after printing and required further thermal curing to reach close to 1000 times modulus contrast[25]. Additionally, the stretchability of the two-stage cured sample was less than 35% in

soft states, which significantly limited the functionalities of the printed structures. Our resin overcomes these limitations by offering much higher modulus contrast and, more importantly, much higher stretchability through a single stage and single ink printing.

Combining highly stretchable elastomers and stiff glassy polymers into one structure can significantly enhance DLP 3D printing for structures and devices that not only require complex geometries but also have integrated heterogeneous properties to meet demanding material requirements in applications such as soft robots, actuators, flexible electronics, biomimetic, pneumatic structures, etc.

### g-DLP printing for composites structures

To verify the capability of the single vat multi-properties g-DLP printing, we print a sample with continuous gradient grayscale from G0 to G70.

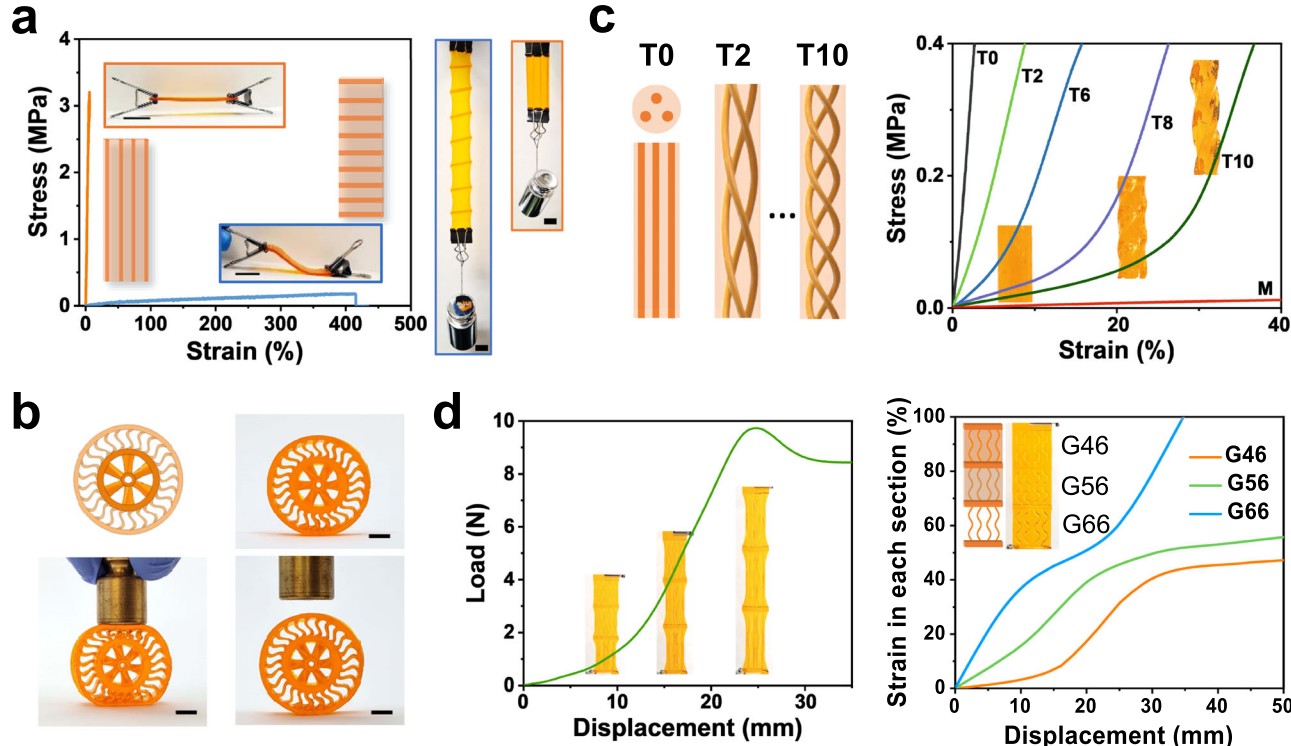

**Fig. 3 | g-DLP printed composite structures. a** Fiber-embedded composites with anisotropic behaviors; **b** Airless tire with rubbery outer circumference and rigid inner hub; **c** Collagen structure with helical fibers with different pitch numbers (the number refers to the turn number of the helical fibers and M refers to the matrix without any fiber); **d** Composite designed with sequential deformation. All scale bar is 1 cm.

The sample is expected to have a continuous Tg transition from ~90 to ~−10 °C. Thus, we exploit the shape memory feature of thermoset polymer to program the sample into a helix shape at 90 °C and fix the shape by rapid cooling in ice water. Afterward, the programmed sample is dropped into an 80 °C water bath for free recovery. The sample exhibits sequential shape recovery from the low Tg region to the high Tg region, which reflects the smooth transition of multi-properties. This experiment is presented in Supplementary Movie 1 and Supplementary Fig. 16. Moreover, we further examine the interface and adhesion between the stiff thermoset and rubbery organogel. As the different curing states are all generated from the same resin, the adhesion is expected to be good. Meanwhile, the inherent transition between different curing states due to diffusion enhances the adhesion between different domains of the g-DLP printed structures. As presented in Supplementary Fig. 17, the energy needs to break the interface between the soft (G60) and stiff (G0) domains is greater than the energy needs to break the soft part itself, demonstrating good adhesion.

Taking advantage of the drastic mechanical property contrast and large stretchability, we first 3D print structures that demonstrate composite-like behaviors. Figure 3a shows a printed composite with stiff fibers (G0) embedded in a flexible matrix (G60), which exhibits a high degree of anisotropy. Perpendicular to the fiber direction, the composite is too soft to hold its own weight (about 1 g), and a 200 g weight stretches the composite more than two times. In the parallel direction, the composite holds the same 200 g weight without observable deformation. Figure 3b shows the prototyping of an airless tire with a compressible rubbery (G60) outer circumference for shock-absorbing. The inner hub structure (G0) is rigid. Such an integrated structure allows the tire to deform when a force is applied in the vertical direction (mimicking a bump on the ground) and to return completely to its original state after the force is removed.

Next, we design composites that mimic artery tissues, whose stress-strain behaviors exhibit a characteristic J-shaped curve[47,48].

Such a J-shape curve is important for the functions of artery tissues in regulating the blood flow: at low pressure, the artery can expand easily to allow more blood flow; as it expands beyond a certain limit, it stiffens to restrict the amount of the flow. This behavior is due to the structure of an artery tissue, which is a soft elastin matrix embedded with stiff and tortuous collagen fibers. At a low stretch of the tissue, the tortuous collagen fibers provide little deformation resistance (due to bending) until they are straightened and become axially stretched at a large tissue stretch ratio[49,50]. Here, we mimic the structure of artery tissue by using the soft elastin matrix (G60) and embedded stiff helical fibers (G0) (Fig. 3c). Cylindrical samples with straight (T0), helical fibers of different pitches (T2, T6, T8, and T10; the digital value refers to the number of turns of the helical fibers), and without fibers (M) are printed. Uniaxial tensile tests show that the printed structures with helical fibers exhibit the J-shape behavior, i.e., it initially has low stiffness, which gradually increases, then has high stiffness, corresponding to the straightening of the stiff fibers. Adjusting the pitch size of fibers (equivalent to tortuosity) can tune the strain range for fiber straightening. For example, T10 shows the most obvious transition (Fig. 3c). These results indicate that g-DLP printing can be used for prototyping artificial tissues that mimic real tissue behaviors.

We can also use multiple grayscale levels to achieve sequential deformation responses under increasing force levels. Figure 3d shows an example of spring-shape stiff fibers (G0) embedded in three sections of a rubbery matrix with different stiffness assigned (G46, G56, and G66) and connected in series, which can achieve the sequential deformation of three sections under the uniaxial tension. The left graphic displays the overall strain change of the entire structure and the right one shows the strain change of each section. A clear time delay of deformation for each section is observed (Supplementary Movie 2), with G46 (the stiffest) showing deformation at a later stage.

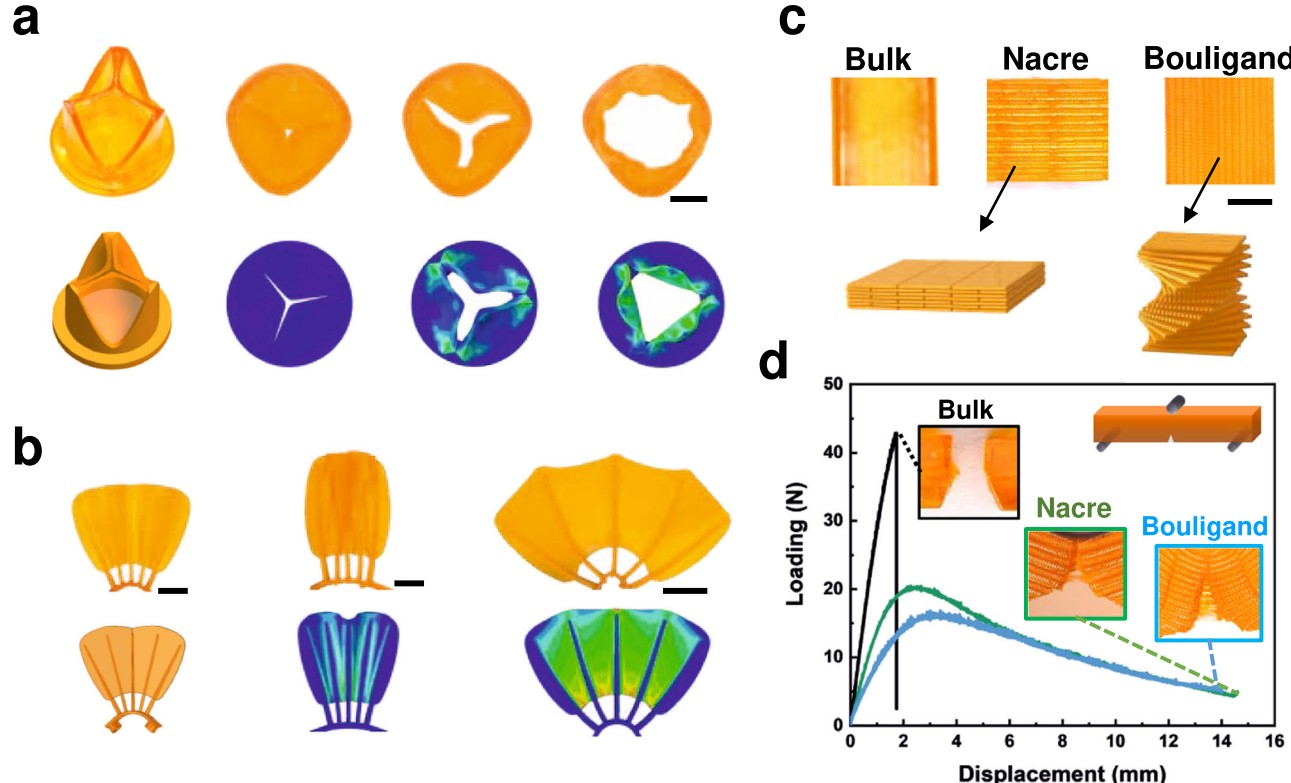

**Fig. 4 | g-DLP printed biomimetic structures. a** Artificial heart valve, scale bar is 1 cm; **b** Fish fin structure, scale bar is 1 cm; **c** Nacre and Bouligand structure, scale bar is 5 mm; **d** Comparison of fracture toughness, insert are samples after test, green line: nacre; blue line: Bouligand.

## g-DLP printing for biomimetic structures

Biomimetic structures consisting of soft and hard phases are difficult to fabricate by conventional DLP methods with a single vat. Here, we exploit the capability of printing such biomimetic structures in a monolithic piece through our single-vat g-DLP printing. Figure 4a shows a printed artificial human heart valve consisting of a rigid support (G0) with three soft valve flaps (G60). The still images in Fig. 4a show the transition between the closed and open states under different fluid flow conditions. The hydrodynamic performance of the g-DLP printed heart valves is presented in Supplementary Movie 3. We also simulate the hydrodynamic performance of the valves by finite element analysis (FEA) using a pressure load on the inside face of the flaps. More details can be found in the SI, Section 2. The simulation results match well with the experiments. This fast, customizable printing method can be an efficient approach to printing patient-specific heart valve models for pre-surgical planning. Figure 4b shows a printed fish fin structure with stiff bony rays (G0) attached to the common fin base (G0) and held by the flexible tissue membrane (G60). Fishes can change the shape of their fins by pushing or pulling the base of the bony rays by muscles and tendons[51,52]. Our printed model can mimic this shape change, as shown in Fig. 4b and Supplementary Movie 4. The fish fin actuation is modeled by FEA using a pressure load normal to the base to cause the opening and closing. The out-of-plane bending of the membrane at the folding state and in-plane bending of the bony rays at expanding state are well-matched with the experimental results. The g-DLP printed fish fin structure accurately replicates such a mechanism, facilitating the rapid biomimetic soft robotic structure fabrication.

We further design biomimetic structures with complex multi-material architectures. Nacre and Bouligand structures are composites that are well known for their intelligent designs to effectively improve fracture toughness by using hard and soft materials[53]. Here, we present the capability of our resin with g-DLP to print structures that exhibit significantly improved fracture toughness by mimicking nacre and Bouligand structures (Fig. 4c). In our design, stiff flakes or fibers (G0) are held by the soft interface (G70) that elongates the crack propagation pathways and dissipates energy. Specific parameters of the structures like flake/fiber size, aspect ratio or detailed arrangement of the two phases, etc., can be tuned to optimize the toughness of the printed composites (Supplementary Figs. 18, 19). Figure 4d shows the three-point bending results of the printed biomimetic structures, with the nacre-like structure having the dimension of 48 mm (L) by 6 mm (W) by 6 mm (t) and 51% of soft materials and the Bouligand structure with the dimension of 56 mm (L) by 6 mm (W) by 6 mm (t) and 48% of soft materials. The three-point bending apparatus has a span of 32 mm. The fracture toughness is 166 and 146 kPa $m^{0.5}$ for the nacre and the Bouligand structure, respectively, which are three to four times higher than the direct cured bulk sample (42 kPa $m^{0.5}$). The g-DLP printing thus demonstrates the capability for concept proofing of various biomimetic structures.

## g-DLP printing of inflatable structures

Next, we demonstrate the rapid fabrication of complex inflatable structures, which have found important applications in soft robots in recent years[54]. The design space for inflatable structures is often limited by the use of only one material. To achieve more complicated shape change using one material, one has to rely on complex geometries[55]. Using multiple materials to achieve complex inflations via 3D printing typically requires multiple-material reservoirs or even a combination of printing technologies[56–58]. Using this resin formulation, stiff inclusions can easily be added anywhere in a soft, stretchable matrix in a sing-vat printing. This opens the door to a wide array of possible inflatable structure designs that cannot be easily achieved using other techniques. Examples of these inflatable structures are presented in Fig. 5. The pressure used in these examples is 30 kPa.

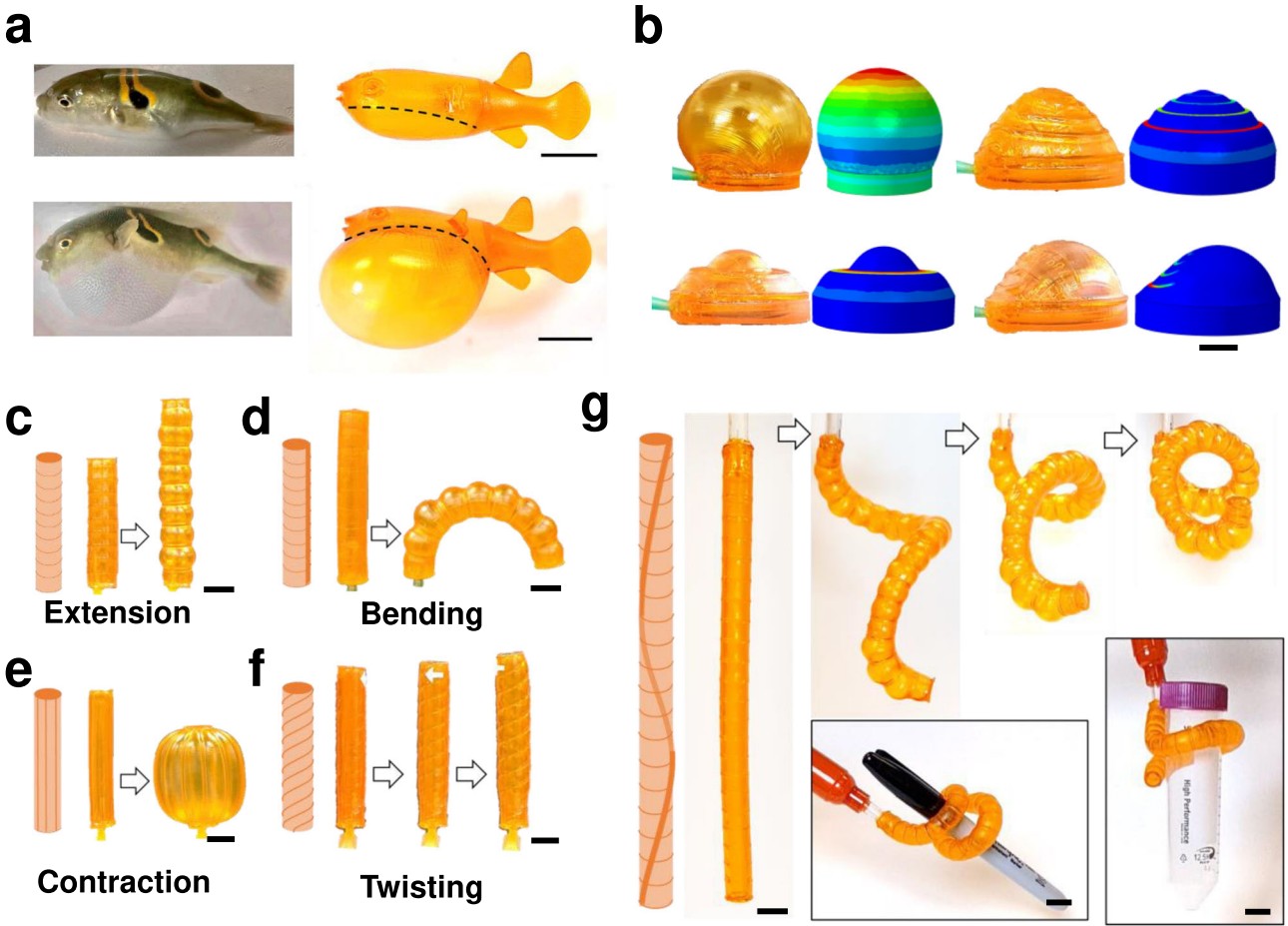

**Fig. 5 | g-DLP printed pneumatic inflatable structures. a** Real puffer fish compared with printed structure; **b** Inflatable membranes with different designs; **c**–**f** Extension, bending, contraction, and twisting actuator; **g** Tentacle-like actuator, inserts are imitating tentacle grabbing. All scale bars are 1 cm.

Figure 5a shows the inflation of a 3D-printed biomimetic pufferfish. The dotted black line represents the material transition region. The parts above that line are printed with G0 so that it is very stiff, while the material below is the much softer G60. When the fish is subjected to internal pressure, the rigid body of the fish undergoes only a small expansion while the soft belly expands significantly (more than 30 times its initial volume; Supplementary Movie 5.), mimicking the behavior of a real pufferfish. This complex biological design can be easily achieved using this resin with g-DLP printing.

Figure 5b shows the actuation of several patterned membrane structures. Each design consists of a shallow rigid cylinder with a rigid cap at the bottom and a soft membrane on the top (the details design can be found in SI, Supplementary Fig. 20, and Supplementary Movie 6). In the top left design, there is no rigid pattern added to the membrane. This represents the default inflation case, which is often the limit of single-material inflatable membranes. However, we can easily add rigid fibers to alter the inflated shape of the membrane. The bottom left panel of Fig. 5b shows a design with a single stiff ring placed in the soft membrane. Internal pressure causes inflation of the soft material between the edge of the cylinder and the outer radius of the stiff ring, which makes the ring pop up. The ring itself does not deform due to its much higher modulus. At the inside of the ring is another stiff membrane, which also pops up due to internal pressure. The top right design features four thin concentric rings. These stiff rings constrain the outward expansion of the membrane, but they do not prevent the expansion of the membrane upward. This causes the final deformation to be more conical in stark contrast to the plain membrane to the left, that takes on a spherical shape. Finally, in the

bottom right panel, the same four concentric one-third circles are printed on the membrane. The left side of the membrane with fibers causes it to take on a more linear shape, while the right side is free to take on the usual spherical shape, leading to asymmetric inflation. The inflation experiment agrees well with the simulations (Supplementary Movie 6). Although we present only four designs here, many other inflatable membrane manipulations can be easily achieved.

Soft pneumatic actuators have been widely used in robotics due to their great adaptability and flexibility than conventional linkage-joint robots[59]. The general approach to fabricating soft actuators is incorporating strain limiters with an airtight soft matrix. Recently reported methods include wrapping inextensible fibers with varied angles, sealing airtight fabric bags, or printing stiff fibers on sealed soft structures[60–62]. But these methods require tedious multi-step manufacturing processes. The g-DLP printing with this resin provides a simple, one-step way to fabricate soft pneumatic actuators with greater adaptability and flexibility than the previous methods. Figure 5c–f shows stiff fibers embedded in a stretchable airtight cylinder matrix that achieve the four basic motions: extension, torsion, contraction, and bending when inflated. Stacking sequences and combinations of those basic motions may give rise to more sophisticated deformation. Figure 5c shows the extension case where stiff rings are placed along the length of a soft tube. Without the rings, the tube would experience mainly an outward deformation when inflated, but the thin rings prevent the outward expansion. Because the stiff rings are not connected, the actuator is much more compliant in the longitudinal direction, causing its length to increase significantly under pressure. Figure 5d is a modification of the extension design

where stiff rings arranged vertically cause an extension of the actuator, but adding a single stiff fiber on one side places an asymmetric constraint on the actuator, leading to a bending motion. Although only a half rotation is shown here, a full rotation can be achieved at a higher pressure. Figure 5e presents the retraction case whose design features stiff fibers that extend straight between the two end caps. Under pressure, the stiff fibers bend outward but resist a change in length. This causes the two end caps to be pulled closer together by the bending fibers. The design in Fig. 5f shows the torsion case, which has two helical rings that spiral between the top and bottom of the actuator. Again, the stiff spirals constrain the outward radial expansion of the tube. The vertical extension is achieved by the unwinding of the spiral structures, which causes the whole actuator to twist, as shown by the attached arrow. Experiments of these actuators are presented in Supplementary Movie 7. The combination of these actuators can be used for complex applications such as soft robotics and biomimicry. Here, we further designed a tentacle-like actuator consisting of bending and twisting, as shown in Fig. 5g. Upon inflation, the actuator gradually wraps into two circles as designed, imitating the tentacle grabbing. The insert shows that it (weight 5 g) can grasp a plastic centrifuge tube (weight 14 g) or a marker (weight 8.5 g) (Fig. 5g inserted), functioning like an elephant trunk. The experiments are presented in Supplementary Movie 8. Compared to the previous design of grippers, which typically require two or three fingers, the gripper enabled by this ink allows a much simple design. We can easily vary the fiber orientation and density to further change the tentacle's inflated shape and motion. This demonstrates this ink's versatility for g-DLP fabrication customized and programmable deforming soft pneumatic actuators.

### g-DLP printing for elastomer sensors

The broad range of properties, from a stretchable rubbery state to a stiff glassy state, is also ideal for the application of flexible sensors with large deformation. Here, we demonstrate the single-vat fabrication of different types of strain sensors via g-DLP printing. The wave-shaped microfluidic channel is printed within the soft matrix and filled with a liquid metal (eutectic gallium–indium alloy (EgaIn, Sigma-Aldrich)). Conductive leads are glued into the ends of the channel for resistance measurement. Figure 6a shows a simple extension strain gauge with a microfluidic channel diameter of 0.8 mm. As the sensor is stretched, the microfluidic channels increase in length and become thinner, dramatically increasing their resistance, which is measured by a multimeter. Figure 6a shows the change in resistance recorded over 24 cycles of stretching to a large deformation (200% strain). Figure 6b shows the design of a pressure sensor with a liquid metal-filled microfluidic channel (0.8 mm) in the soft membrane. When internal pressure is applied, the membrane is deformed upwards, increasing the resistance in the liquid metal. The rest of the sensor is printed at a higher grayscale (G0) so that deformation is confined to the membrane alone. It can accurately capture the different pressure levels with relatively small deformation, as shown in Fig. 6b. A unique feature of the proposed grayscale resin is that the membrane's stiffness can be adjusted to alter the sensitivity of the sensor without the need to change the thickness.

The g-DLP 3D printing with this ink also has unique advantages for fabricating customized human wearable electronic devices and sensors. The rubbery state covers the mechanical pliability range of human skin (Young's modulus ≈130–657 kPa)[49]. Also, the stiff state enables intimate conformability to mount the body without extra fixtures. Figure 6c shows a g-DLP-printed finger-mounted sensor for human-machine interfaces. The smaller microfluidic channel (0.4 mm) results in a higher sensitivity that can detect a small bending of the finger, as shown in Fig. 6c. Similar sensors can easily be designed to fit different users or joints for both humans and robots. Overall, these examples demonstrate the capability for fabricating challenging designs for various sensor applications.

## Discussion

Combining materials with highly contrasted properties, from stretchable elastomers to hard thermosets in 3D printing enables the fabrication of highly complex 3D functional structures. Soft and stretchable gel materials consisting of polymeric networks with various solvents, including hydrogel (water), ionogel (ionoic liquid), and organogel (organic solvent), have been widely reported for 3D printing. These gels generally rely on hydrogen bonding between solvent molecules and polymers. Here, the g-DLP ink formulation is inspired by these gels. At low conversion, monomers are able to perform like solvent molecules in organogel via hydrogen bonding, while at high conversion, monomers crosslink into a stiff thermoset. The UV-curable AUD oligomers and 2-HEA monomers are both favorable to form hydrogen bonding resulting in an elastic and stretchable organogel at low conversion. Additionally, monomer IBOA ensured the stiffness of the hard thermoset at high conversion. The g-DLP printing with this rationally designed ink provides the simple single vat approach to realize the multi-properties printing of stretchability and stiffness from both states. Besides the used resin in this work, more UV-curable resin can be designed with the same strategy that uncured monomer can interact with the crosslinked network through not only hydrogen bonding but also ion-dipole or π−π stacking interaction to form a stretchable gel at low DoC (an exemplary resin formulation is presented in Supplementary Fig. 21).

The current ink does not require revising the DLP printing and thus retains all the advantages offered by the DLP technique. For example, multiple objectives can be printed in the same batch at a high printing speed (1 mm/min). Supplementary Fig. 22 shows that 12 tentacle-like actuators can be printed simultaneously. This can greatly increase the production rate.

Despite the significant advantages of single vat g-DLP printing, some limitations still need to be addressed further. Firstly, the g-DLP printed rubbery part is an organogel with liquid phase (uncured monomers) immobilized in the hydrogen bonding dominated gel network. Just like general organogels, it is sensitive to high temperatures since it weakens the hydrogen bonding and makes the liquid phase volatile. Secondly, we observe between the interface of the stiff part and the rubbery part, there is a pixel scale transition layer (50–100 μm, Supplementary Fig. 23), which limits the g-DLP printing feasibility for micron-scale structures. Although such a transition layer is negligible for all the demonstrated printing size scales in this work, it could be optimized further with different approaches like using a projector with smaller pixel size, algorithm-corrected grayscaled images, or resin with more photo absorber[63]. Moreover, the low grayscale will result in size distortion on the x-y plane, which is a general problem facing g-DLP printing and will affect the printing resolution for micro-structures. Using an increased projecting area for lower grayscale (Supplementary Fig. 24) or developing computational models can address the challenge. But finding a more general solution to this limitation is beyond the scope of this work. Last, the single-vat grayscale cured samples have varying conversion degrees for multi-properties. Consequently, those uncured acrylate groups make the printed structure UV-sensitive. Further exposure under UV will reduce the property contrast until it becomes a uniformly stiff structure. Our previous work has demonstrated that using a second stage curing through amine-acrylate Michael addition reaction could consume the unreacted acrylate groups[64]. Polyamine with long and flexible backbones can be used to compose a hybrid two-stage resin, which is part of our following work. Nevertheless, single vat printing with instantly multi-properties function capability has its unique advantage and values for many application scenarios.

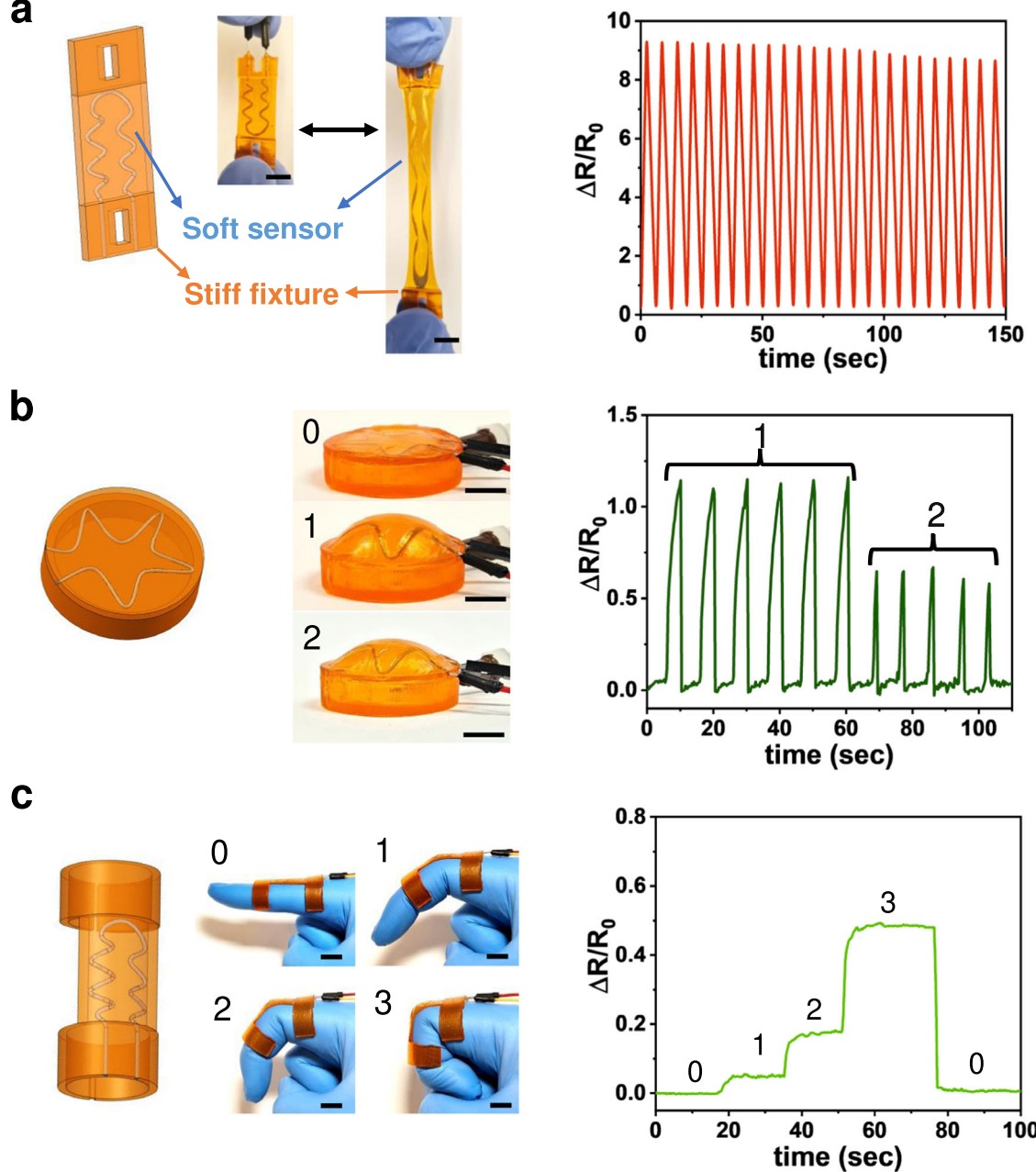

**Fig. 6 | g-DLP printed sensor devices. a** Strain sensor and ΔR/R0 response of 200% strain; **b** Pressure sensor and ΔR/R0 response with corresponding pressure; **c** Finger sensor and ΔR/R0 response with corresponding bending. All scale bar is 1 cm.

In summary, a resin is developed for grayscale digital light processing (g-DLP) printing to fabricate parts with drastically varying properties in a single pot resin. This resin offers the capability of continuously varying modulus from 0.016 to 478 MPa, nearly 30,000 times difference. Also, the material printed with low-intensity light is highly stretchable (~1500%). Using the DLP technique also permits high printing speed (1 mm/min). The ability to fabricate parts with very rigid regions and flexible regions all at once and at high speed is a crucial advantage in rapid prototyping applications. The resin formula presented here possesses significant advantages over previously reported DLP resins, which allows for unprecedented freedom in printable designs. With such a wide range of properties, we demonstrate robust and adaptable capability for various applications requiring multiple properties in a monolithic structure. It provides a rapid manufacturing approach for prototyping and performance validation, which can be used for soft robots and actuators, metamaterials, flexible electronics, biomimetic, and pneumatic structures.

## Methods

### Ink preparation
The photocurable resin was prepared by mixing 2-hydroxyethyl acrylate (Sigma-Aldrich, MO, USA), isobornyl acrylate (Sigma-Aldrich), and AUD (Ebecryl 8413, Allnex, GA, USA) with the weight ratio of 20:60:20. Then 1 wt% photoinitiator (Irgacure 819, Sigma-Aldrich) and 0.05 wt% photo absorber (Sudan I, Sigma-Aldrich) are added.

### Digital light processing 3D printing
3D printing is performed with a bottom-up DLP printer that employs a 385 nm UV-LED light projector (PRO4500, Wintech Digital Systems Technology Corp., Carlsbad, CA, USA) and a linear translation stage

(LTS150 Thorlabs, Newton, NJ, USA). A homemade container with an oxygen-permeable window (Teflon AF-2400, Biogeneral Inc., CA, USA) is used as the resin vat. The designed 3D structures are sliced into image files with a thickness of 0.05 mm and then converted into grayscaled image files with a MATLAB script. The continuous liquid interface production (CLIP) approach is utilized at the optimized speed of 3 s/layer to print the designed 3D structures. All samples are raised with isopropyl alcohol after printing and wrapped in aluminum foil for further testing or demonstration. The light intensity of the printer is calibrated with a photometer (ILT1400-A Radiometer, International Light Technologies Inc., MA, USA) before printing.

## Characterizations

The uniaxial tension tests are performed with a universal test machine (Insight 10, MTS Systems Corp., Eden Prairie, MN, USA) with a cross-head speed of 5 mm/min. Dynamic thermomechanical properties are conducted on a DMA machine (Q800, TA Instruments, New Castle, DE, USA) with a temperature ramped at a rate of 10 °C/min. The degree of curing is decided by normalized FTIR (Nicolet iS50 spectrometer, Thermo Fisher Scientific, Waltham, MA, USA) peak intensity of the acrylate group present at 809 cm$^{-1}$. Multiple tests are conducted for each sample to guarantee reproducibility.

## Data availability

The authors declare that the data supporting the findings of this study are available within the article and its Supplementary Information file. Raw data of the resin properties generated in this study are provided in the Source Data file with this paper. All other data underlying the results, including the digital designs of those printed structures, are available from the corresponding author upon request. Source data are provided with this paper.

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

## Acknowledgements

H.J.Q. would like to acknowledge the support from an AFOSR grant (FA9550-20-1-0306; Dr. B.-L. "Les" Lee, Program Manager), Toyota North America, the gift funds from HP, Inc. and Northrop Grumman Corporation. This work is performed in part at the Georgia Tech Institute for Electronics and Nanotechnology, a member of the National Nanotechnology Coordinated Infrastructure, which is supported by the National Science Foundation (ECCS-1542174).

## Author contributions

L.Yue and H.J.Q. conceived the concept. L.Yue, H.J.Q., Y.S., M.T., and T.N. designed the experiments. S.M.M. performed the FEA simulations. L.Y., S.M.M., and X.S. conducted the composite structures design. L.Y. performed the property characterization. L.Yue and H.J.Q. prepared the manuscript. All authors discussed the results and commented on the paper.

## Competing interests

L.Yue, H.J.Q., Y.S., and M.T. are coinventors on two provisional patent applications. (1) Resins for digital light processing 3d printing (no. 17/833,144, filed 6 June 2022); (2) digital light processing 3d-printed monolithic substrates with integrated and embedded sensors (no. 17/977,314, filed 31 October 2022), filed by Toyota Motor Engineering and Manufacturing North America Inc. and Georgia Tech Research Corporation. The authors declare that they have no other competing interests.
