## [Peer Review File · Nature Communications]

Single-Vat Single-Cure Grayscale Digital Light Processing 3D Printing of Materials with Large Property Difference and High StretchabilityReviewers' Comments:

Reviewer #1:

Remarks to the Author:

Please see the attachment.

The challenge facing property modulation via light-based, grayscale 3D-printing is twofold: *i*) to be able to modulate a property of interest and *ii*) to create a large and preservable internal gradient of that property. Yue et al. present a smart photoresin design that mimics the stretchability of an organogel at moderate conversion and attains the stiffness of a thermoset at high conversion. Uniquely, the extensive hydrogen bonding that holds linear chains together at its rubbery state is introduced deliberately by the authors to ensure that a workpiece is highly stretchable when soft. The authors further demonstrated that enabling the co-manufacturing of both stretchable and stiff elements within a monolith has important implications for fast prototyping functional 3D structures using the CLIP technique. The manuscript is sound and inspiring, and contributes to addressing both challenges above. The ink allows a user to choose between stiffness and stretchability, and the authors show convincingly that large property contrast could be generated, and preserved at least temporarily. The following comments are meant to help further improve the impressive work.

Generating and preserving property contrast

A general strategy of grayscale additive manufacturing is to carefully regulate dose of illumination to arrest the degree of cure (DoC), which can be leveraged to drive distinct phase change via, e.g., shifting the glass transition temperature relative to the ambient environment. The largest achievable modulus contrast is always capped by the highest DoC within the curing time per slice. In this work, the achieved stiffness, without additional postprocessing, is comparable to previously published works that would typically require multi-stage curing (Fig. 2f). This elegance in itself poses a dilemma – the advantage of the new, rationally designed ink over existing ones would be most distinctive in a single-vat, single-cure context, but preserving the achieved property gradient would almost certainly require an additional stage of postprocessing. A significant contribution by the authors is the great stretchability achieved when the curing is curtailed to a rubbery state. In the second half of the manuscript, the majority of the demonstrated functional structures rely on the co-existence of both glassy and rubbery states within a monolithic construct to function, which, again, highlights the concern regarding the transient nature of the internal property contrast as well as the great challenges in preserving such contrast under less stringent conditions (e.g., in the presence of UV).

Multimaterial constructs

Roughly 48% of the main text (2660/5539 words, excluding M&M) is devoted to an elaboration of how the new ink works with gDLP to quickly 3D-print wittfully chosen constructs that combine stiff and elastic elements to serve novel functionalities. While the unique advantage of the photoresin formulation is appreciated, it might be beneficial to delineate the contribution of smart mechanical design from that of the novel ink property. I'm not confident in commenting on the uniqueness of the ink in prototyping these demonstrative structures – whether they have been manufactured using alternative techniques or does the new ink introduce performance upgrades. I believe readers will better appreciate the authors' work if such information could be briefly summarized in the Introduction or before each construct is presented. Also, it would be helpful to state explicitly how the challenges in postprocessing (e.g., the probability to preserve the internal stretchability-stiffness contrast) may or may not impair these applications.

Orthogonality

From a 3D-printing perspective, the work would be truly groundbreaking if emphasis is put on demonstrating two key orthogonality.

Geometry control vs. Property modulation

In binary DLP-based AM, geometric accuracy is improved by calibrating unit feature with regard to dose of illumination. Although the authors have candidly revealed the “distortion” (Fig. S20, I do not think the dot features are distorted, but scaled) stemming from variation in gray level, it's not straightforward to me how this interdependence can be compensated for when a more complex geometry with internal property gradients needs to be printed. If geometry control and property modulation are not orthogonal, use of the new ink in gDLP will be limited to structures consisting of fairly simple 2D slices, with only a small number of discretely graded properties (instead of a

“continuously” transition of material properties). This limitation has been apparent in the demonstration of functional constructs.

Stiffness vs. Stretchability

The modulation of stiffness and stretchability is not orthogonal, i.e., one would not be able to print objects that are both stiff and elastic, or both soft and brittle. This is not a disadvantage but ought to be stated upfront for a better expectation management. Within each state (rubbery/glassy), there’s room for continuously tuning a given property via greyscale illumination, but the continuity does not extend beyond the glass transition. I’d recommend that the authors modify the (somehow misleading) shape of the green ellipse in Fig. 2f because it does not reflect the inherent discontinuity between the stretchable and stiff states. It also does not signify that the true novelty of the rationally designed ink is to enable, by regulating DoC, the integration of both stiff and highly stretchable components via single curing. On a related note, information on the transitional behavior of the ink between the rubbery and the glassy states is scarce. Such information would help readers understand the formation of potential artefacts (in both the X-Y direction and between vertical slices) and evaluate the reproducibility of the results.

Miscellaneous

1. Briefly surveying previous efforts of property modulation via single-step dose regulation (gDLP) in the introduction (esp. from outside Jerry’s group) would be appreciated.
2. It is not clear why FEA is needed – it provides very limited extra support for the claims and scrutinizing the technical details in the SI invites questions regarding parameterization.
3. Could the authors comment on if the ink only suits conventional AM?
4. Pg. 2: “number of available material properties...” Please double check. I believe the orthogonal chemistry employed in a few 3D-printing studies (including Ref.9) allows unlimited number of grades (i.e., continuous gradients) with only two light sources of different wavelengths.
5. Pg. 18 “multi-step”; Pg. 24 “explosion”?
6. Pg. 23 “... retains all the advantages offered by the DLP technique.” The ink may also inherit some disadvantages of DLP-based vat photopolymerization, such as embedding lamination-induced artefacts between slices. The author mentioned that there would be pixel scale transition between stiff and rubbery parts (an image would be appreciated). Will such transition complicate the lamination, i.e., what does the interface between two vertical slices look like if one of them is rubbery and the other glassy?

Additional work

The manuscript is technically sound and requires little additional work to be publishable in a specialized journal. A few extra tasks may help the readership better appreciate its novelty.

- a. The authors’ claim that their work represents a general strategy of rationally designing novel inks may be better supported by including more than one exemplary resin formulation.
- b. The authors may consider naming a few applications (other than fast-prototyping) that are not susceptible to the difficulty of preserving internal property contrast over a longer term.
- c. The mechanical properties of most demonstrations are binary (stiff + elastic), with the exception of the sequential deformation designs (ternary, elastic). The authors’ claim that the ink will help obtain a continuously varying modulus (Pg. 24) may be better supported by presenting an example with a smooth internal property gradient. However, this may require a better understanding of the glass transition behavior?
- d. Identifying the dose-DoC correlation and in particular the dose threshold for glass transition in terms of energy per unit ink volume (instead of greyscale%) will definitely help readers reproduce the results.

Reviewer #2:

Remarks to the Author:

The authors describe a straight forward approach to fabricating materials that possess controllable stiffness using greyscale 3D printing. Although the work uses common monomers and crosslinkers, the combination of precursors used is unique and the formulation yields materials with tunable stiffnesses based on the light intensity (degree of polymer conversion). The authors are able to demonstrate high contrast in material stiffness, beyond what has previously been demonstrated due to an extensive network of hydrogen bonding. With this new ability to spatially control material stiffness with high contrast in 3D objects, the group demonstrate a number of applications, including airless tires, strain sensors, biomimetic arteries, heart valve, nacre, fish fin, pneumatic actuators. The authors are careful not to oversell the work and highlight some of the challenges, such as the presence of unreacted acrylate groups that lead to the materials UV-sensitivity, layer artifacts and the limitation in applications due to print resolution. The authors offer some potential solutions to these challenges.

Comments:

1. The formulation contains 20% w/w of urethane crosslinker. This is a very low amount of crosslinker for a 3D printable formulation. The network of hydrogen bonding presumably helps in providing structural integrity to the materials for it to be printable. At a greyscale of 60, the polymer has a polymer conversion of ~60%. Given the polymerization reaction is chain growth, and the above low crosslinker concentrations, there is a high concentration of unreacted monomer and/or short oligomers. The isobornyl acrylate has a relatively low boiling point and does not hydrogen bond and would thus be susceptible to becoming volatile. Can the authors comment on the volatility or even the loss of materials through handling the printed objects given that the materials is likely made up low molecular weight species.
2. Can data points corresponding to this work be included in Figure 2f?
3. Can these work be included in Figure 2f
 - Tingting Zhao, Ran Yu, Shan Li, Xinpan Li, Ying Zhang, Xin Yang, Xiaojuan Zhao, Chen Wang, Zhichao Liu, Rui Dou, and Wei Huang ACS Applied Materials & Interfaces 2019 11 (15), 14391-14398 DOI: 10.1021/acsami.9b03156.
4. Correct the typos in lines 424- 427. These gels generally rely on hydrogen bonding between solvent molecules and polymers. Here, the g-DLP ink formulation is inspired by these gels. At low conversion, monomers are able to perform like solvent molecules in organogel via hydrogen bonding, while at high conversion, monomers crosslink into a stiff thermoset.
5. Can the authors add a description of how the objects are processed/cleaned post-printing. Given point #1, this may influence the properties of the objects.

Reviewer #3:

Remarks to the Author:

Qi and co-workers report on the fabrication of multi-material 3D structures using grey scale stereolithography. By careful selection of the acrylate-based monomers, they were able to vary the mechanical properties over a broad range by printing with a single vat. In contrast to their previous work, the newly developed material does not require a thermal post-curing step and benefits from a distinctive extension of the elongation. Thus, in my opinion the topic of the manuscript makes a significant contribution to the field and will be of interest for a broader scientific community as multi-material structures are relevant for numerous research fields. This was also demonstrated by the authors, who printed multi-material 3D structures applicable for soft robots/actuators or electronics. Suggested revisions are listed in the following:

- (1) The authors showed that the printed samples retain their mechanical properties over several days at room temperature in a light protected environment. Along with light, the partly cured networks are also expected to be sensitive to higher temperature. Did the authors study if higher temperatures (> 70 °C) affect the multi-material properties?
- (2) The authors comprehensively discuss the limitations of their system in the manuscript. Regarding

the resolution, they mention a pixel scale transition layer of 50 – 100 μm , which limits the resolution of the developed resin. However, I am missing the actual resolution of the multi-material domains in the manuscript and I recommend that the authors address this point in their work.

(3) Whilst the transition layer negatively affects the resolution, it might contribute to a higher adhesion strength between the soft and the hard part of the printed structures. The authors should address and comment on the interlayer adhesion between the soft and rigid domains, which is often a crucial issue in 3D printed multi-material structures.

(4) In the experimental part, the reference resins are missing.

Responses to Reviewers

Reviewer #1

General comments:

The challenge facing property modulation via light-based, grayscale 3D-printing is twofold: i) to be able to modulate a property of interest and ii) to create a large and preservable internal gradient of that property. Yue et al. present a smart photo resin design that mimics the stretchability of an organogel at moderate conversion and attains the stiffness of a thermoset at high conversion. Uniquely, the extensive hydrogen bonding that holds linear chains together at its rubbery state is introduced deliberately by the authors to ensure that a workpiece is highly stretchable when soft. The authors further demonstrated that enabling the co-manufacturing of both stretchable and stiff elements within a monolith has important implications for fast prototyping functional 3D structures using the CLIP technique. The manuscript is sound and inspiring, and contributes to addressing both challenges above. The ink allows a user to choose between stiffness and stretchability, and the authors show convincingly that large property contrast could be generated, and preserved at least temporarily. The following comments are meant to help further improve the impressive work.

Response: We appreciate the reviewer for acknowledging the novelty of our work and confirming the contribution of our work on the two critical challenges in DLP 3D printing. We would like to thank the reviewer for taking the time to carefully review our manuscript and providing constructive comments and suggestions. In the following, we provided point-by-point responses of the specific comments.

1. **Comments on:** Generating and preserving property contrast

Comments #1.1: A general strategy of grayscale additive manufacturing is to carefully regulate dose of illumination to arrest the degree of cure (DoC), which can be leveraged to drive distinct phase change via, e.g., shifting the glass transition temperature relative to the ambient environment. The largest achievable modulus contrast is always capped

by the highest DoC within the curing time per slice. In this work, the achieved stiffness, without additional postprocessing, is comparable to previously published works that would typically require multi-stage curing (Fig. 2f). This elegance in itself poses a dilemma – the advantage of the new, rationally designed ink over existing ones would be most distinctive in a single vat, single-cure context, but preserving the achieved property gradient would almost certainly require an additional stage of postprocessing. A significant contribution by the authors is the great stretchability achieved when the curing is curtailed to a rubbery state. In the second half of the manuscript, the majority of the demonstrated functional structures rely on the co-existence of both glassy and rubbery states within a monolithic construct to function, which, again, highlights the concern regarding the transient nature of the internal property contrast as well as the great challenges in preserving such contrast under less stringent conditions (e.g., in the presence of UV).

Response: We thank the reviewer for the positive feedback on our work and for pointing out the elegance and its dilemma of the g-DLP 3D printing. We agree with the reviewer's comment on grayscale printing. The most significant advantage for this work is to create instantly multi-properties capability in single-vat and single-cure printing to overcome the twofold challenges as the reviewer mentioned in general comments. It has practical value for certain scenarios as we discussed in the manuscript. We are continually studying to overcome this dilemma of g-DLP 3D printing, and hopefully this work could inspire the scientific and industrial community to explore a better solution for rapid multi-materials 3D printing.

2. Comments on: Multimaterial constructs

Comments #2.1: Roughly 48% of the main text (2660/5539 words, excluding M&M) is devoted to an elaboration of how the new ink works with gDLP to quickly 3D-print wittfully chosen constructs that combine stiff and elastic elements to serve novel functionalities. While the unique advantage of the photo resin formulation is appreciated, it might be beneficial to delineate the contribution of smart mechanical design from that of the novel ink property. I'm not confident in commenting on the

uniqueness of the ink in prototyping these demonstrative structures – whether they have been manufactured using alternative techniques or does the new ink introduce performance upgrades. I believe readers will better appreciate the authors' work if such information could be briefly summarized in the Introduction or before each construct is presented. Also, it would be helpful to state explicitly how the challenges in postprocessing (e.g., the probability to preserve the internal stretchability-stiffness contrast) may or may not impair these applications.

Response: We thank the reviewer for bringing up this valuable suggestion. The g-DLP 3D print with the designed ink provides a simple and efficient platform to fabricate the structures (which were difficult or impossible to do with conventional AM) we demonstrated in the manuscript. Taking the tentacle-like actuator as an example, the design has two groups of stiff fibers embedded the soft and thin wall of the tube. Fabricating such actuator using conventional methods (including AM assisted method) would be extremely challenging (if not impossible). For instance, a typical method used in literature is to print the tube with cavities and then fill the cavities with a second stiff material. However, this approach cannot be used in the current design as the circular rings cannot be filled by other materials as they are close structures. In addition, the wall thickness is about 1 mm, making it very challenging to print cavities in the wall. We have followed the reviewer's suggestion and added related information about the previous manufacturing techniques of those structures accordingly. We also updated the discussion about postprocessing in the discussion section to help the readers comprehensively understand the possible challenges.

We added the following sentences in the second paragraph of the Introduction section (highlighted on pages 3-4):

“For example, DIW printing was used to write stiff fibers on prefabricated (by molding or 3D printing) soft elastomer structures^{17,22}. Previously reported programmable curvatures of tube-like actuators with multimaterials were fabricated similarly by using multi-step processing with different techniques. Overall, the multi-ink or multi-method AM techniques are often not efficient and have limited capability to fabricate structures with multimaterial properties with complicated distributions.”

We added the following sentences in the g-DLP printing of inflatable structures section on page 19:

“The general approach to fabricating soft actuators is incorporating strain limiters with an airtight soft matrix. Recently reported methods include wrapping inextensible fibers with varied angles, sealing airtight fabric bags or printing stiff fibers on sealed soft structures”

Also we added the following sentence in the materials and methods section on pge 27:

"All samples were raised with isopropyl alcohol after printing and wrapped in aluminum foil for further testing or demonstration."

3. Comments on: Orthogonality

Comments #3.1: From a 3D-printing perspective, the work would be truly groundbreaking if emphasis is put on demonstrating two key orthogonality.

Response: We thank the reviewer for these constructive comments and suggestions on orthogonality that could help improve the quality of the manuscript.

Comments #3.2: Geometry control vs. Property modulation.

In binary DLP-based AM, geometric accuracy is improved by calibrating unit feature with regard to dose of illumination. Although the authors have candidly revealed the "distortion" (Fig. S20, I do not think the dot features are distorted, but scaled) stemming from variation in gray level, it's not straightforward to me how this interdependence can be compensated for when a more complex geometry with internal property gradients needs to be printed. If geometry control and property modulation are not orthogonal, use of the new ink in gDLP will be limited to structures consisting of fairly simple 2D slices, with only a small number of discretely graded properties (instead of a2 "continuously" transition of material properties). This limitation has been apparent in the demonstration of functional constructs.

Response: The reviewer brings up a good point that the final geometry is somewhat dependent on which grayscale values are used for printing. In the current work, a manual compensation is used to achieve the desired geometry. For example, we used

an increased projection area for lower grayscale (Fig. R1; Supplementary Fig. S24) but this method could be difficult to perform on more complicated designs with continuous gradients. We should note that this is a general problem facing g-DLP printing, not something that is unique to the resin described herein [1-4]. Several recent works have attempted more general solutions to this problem [5, 6]. In addition, we are currently working on developing computational models to address these challenges. Finding a more general solution to this limitation is beyond the scope of this work. We have also observed that the size difference is much less apparent at larger feature scales.

We added the following sentences in the discussion section on page 25:

“Using an increased projecting area for lower grayscale (Supplementary Fig. S24) or developing computational models can address the challenge. But finding a more general solution to this limitation is beyond the scope of this work.”

Figure R1. (Figure S24) g-DLP printing could result a size distortion on the printing surface, which could decrease the resolution of small structures. Adjust the projecting size could compensate such distortion. a) Projection image of designed cylinders from G0 to G70; b) projection images of designed cylinders with compensation; c) size and heights of printed cylinders; d) size of printed cylinders with compensation. All scale bars are 500 μm .

1. Montgomery, S.M., et al., *A reaction–diffusion model for grayscale digital light processing 3D printing*. *Extreme Mechanics Letters*, 2022. **53**: p. 101714.
2. Montgomery, S.M., et al., *The 3D printing and modeling of functionally graded*

- Kelvin foams for controlling crushing performance*. Extreme Mechanics Letters, 2021. **46**: p. 101323.
3. Zhang, Y.-F., et al., *Miniature Pneumatic Actuators for Soft Robots by High-Resolution Multimaterial 3D Printing*. Advanced Materials Technologies, 2019. **4**(10): p. 1900427.
 4. Wang, Y., D. Xue, and D. Mei, *Projection-Based Continuous 3D Printing Process With the Grayscale Display Method*. Journal of Manufacturing Science and Engineering, 2019. **142**(2).
 5. Zhou, C., Y. Chen, and R.A. Waltz, *Optimized Mask Image Projection for Solid Freeform Fabrication*. Journal of Manufacturing Science and Engineering, 2009. **131**(6): p. 061004-1.
 6. You, S., et al., *Mitigating Scattering Effects in Light-Based Three-Dimensional Printing Using Machine Learning*. Journal of Manufacturing Science and Engineering, 2020. **142**(8): p. 081002.

Comments #3.3: Stiffness vs. Stretchability

The modulation of stiffness and stretchability is not orthogonal, i.e., one would not be able to print objects that are both stiff and elastic, or both soft and brittle. This is not a disadvantage but ought to be stated upfront for a better expectation management. Within each state (rubbery/glassy), there's room for continuously tuning a given property via grayscale illumination, but the continuity does not extend beyond the glass transition. I'd recommend that the authors modify the (somehow misleading) shape of the green ellipse in Fig. 2f because it does not reflect the inherent discontinuity between the stretchable and stiff states. It also does not signify that the true novelty of the rationally designed ink is to enable, by regulating DoC, the integration of both stiff and highly stretchable components via single curing. On a related note, information on the transitional behavior of the ink between the rubbery and the glassy states is scarce. Such information would help readers understand the formation of potential artefacts (in both the X-Y direction and between vertical slices) and evaluate the reproducibility of the results.

Response: We thank the reviewer for bringing up another very good point about the properties transition at the threshold. We agree with the reviewer, the transition between the organogel and stiff thermoset state is too sensitive to get continuous results. We followed the reviewer's suggestion and updated Figure 2f by Figure R2:

Figure R2. The updated Figure 2f) a comparison of the mechanical property range between the single vat g-DLP printing with generally reported or commercially available DLP ink.

We also added the following sentences on page 10:

"It was also notable that the transition between the stiff thermoset and rubbery gel state was not continuous in this plot. This was because the printing properties were too sensitive to control at this threshold DoC range."

4. Comments on: **Miscellaneous**

Comments #4.1: Briefly surveying previous efforts of property modulation via single-step dose regulation (gDLP) in the introduction (esp. from outside Jerry's group) would be appreciated.

Response: We thank the reviewer for the suggestion. In the revision, we have added the following works about single step g-DLP (outside of our group) in the introduction.

27. Peterson, G. I. et al. Production of Materials with Spatially-Controlled Cross-Link

Density via Vat Photopolymerization. *Acs Appl Mater Inter* 8, 29037-29043

29. Zhang, Z., Corrigan, N., Bagheri, A., Jin, J. & Boyer, C. A Versatile 3D and 4D Printing System through Photocontrolled RAFT Polymerization. *Angewandte Chemie International Edition* 58, 17954-17963 (2019).

30. Valizadeh, I., Al Aboud, A., Dörsam, E. & Weeger, O. Tailoring of functionally graded hyperelastic materials via grayscale mask stereolithography 3D printing. *Additive Manufacturing* 47, 102108 (2021).

Comments #4.2: It is not clear why FEA is needed – it provides very limited extra support for the claims and scrutinizing the technical details in the SI invites questions regarding parameterization.

Response: The FEA may not be needed to support the claims of this work, but we believe that having a predictive model for a material with such wide tunability is helpful to reduce iteration during the design process. In cases where a target deformation is desired, for example, using a predictive model can allow for a much faster, more accurate design of the multi-material distribution [1-3]. In this work, we do not use the FEA model in such a predictive capacity, but we still believe it is informative to include examples of a working numerical model.

We agree that there was a lack of clarity in the SI regarding how the parameters were chosen, so we have made the following clarifications in the SI regarding the material properties for each scenario:

"The values of 300kPa and 500MPa are chosen to represent the materials G70 and G40, which are used for printing."

"In the fish fin analysis, the stiff and soft regions are assigned Young's moduli of 500MPa and 100kPa, respectively, and all regions are meshed with C3D10 tetrahedral elements. The value of the G70 material is reduced to 100kPa for the fish fin simulation to represent the lower effective light dose received due to the thinness of the soft layer. In the heart valve analysis, the stiff and soft regions are assigned Young's moduli of 500MPa and 300kPa, respectively. The properties for the heart valve represent the grayscale values of G0 and G70, which were used for printing."

1. Montgomery, S.M., et al., *The 3D printing and modeling of functionally graded Kelvin foams for controlling crushing performance*. Extreme Mechanics Letters, 2021. **46**: p. 101323.
2. Zhang, Q., et al., *Shape-Memory Balloon Structures by Pneumatic Multi-material 4D Printing*. Advanced Functional Materials, 2021. **31**(21): p. 2010872.
3. Huang, L., et al., *Ultrafast Digital Printing toward 4D Shape Changing Materials*. Advanced Materials, 2017. **29**(7): p. 1605390.

Comments #4.3: Could the authors comment on if the ink only suits conventional AM?

Response: Yes, it's able to use with conventional DLP AM. Since the mechanism of g-DLP printing is localized modulating the light does to tune the DoC, conventional DLP printer should be also capable to print with this ink. For printers that cannot project grayscale images, one can modulate the exposure time to tune the DoC.

We also address this comment in the discussion section on page 24:

“The current ink does not require revising the DLP printing and thus retains all the advantages offered by the DLP technique.”

Comments #4.4: Pg. 2: "number of available material properties..." Please double check. I believe the orthogonal chemistry employed in a few 3D-printing studies (including Ref.9) allows unlimited number of grades (i.e., continuous gradients) with only two light sources of different wavelengths.

Response: We thank the reviewer for this comment. Ref. 9 with the orthogonal chemistry provided very limited continuous gradients for each resin, for example they can modulate the stress at 30% tensile strain from 0.07 to 0.19MPa (by UV light) or 0.02 to 0.04MPa (by visible light) from a single vat printing. We revised the related sentences in the Introduction section on page 4:

"In addition, the number of available material properties in multi-vat approaches is

limited by the number of vat, making it hard to obtain a continuous transition of material properties."

"However, the quality of printed parts relies on the alignment of two light sources and the tunable property range is limited in a single vat."

Comments #4.5: Pg. 18 "multi-step"; Pg. 24 "explosion"?

Response: We thank the reviewer for catching this typo. We have revised the content accordingly and carefully checked the manuscript to ensure the quality of the writing.

Comments #4.6: Pg. 23 "... retains all the advantages offered by the DLP technique." The ink may also inherit some disadvantages of DLP-based vat photopolymerization, such as embedding lamination-induced artefacts between slices. The author mentioned that there would be pixel scale transition between stiff and rubbery parts (an image would be appreciated). Will such transition complicate the lamination, i.e., what does the interface between two vertical slices look like if one of them is rubbery and the other glassy?

Response: We thank the reviewer for this good suggestion. We have conducted additional to create laminated samples and added the following optical microscope images in supporting information as Figure S23 (Figure R3 below). G0 and G60 laminated samples with a layer thickness of 200 μm were printed on two directions as shown in the Figure R3. The G0/G60 interface was more clear on the vertical direction with a thinner transition less than 50 μm (1pixel) as shown in Figure R3a. However, the diffusion-induced transition was more clear on the horizontal direction(Figure R3b). The transition thickness was around 50 to 100 μm (1-2 pixel). Using more photo absorber can help increase the printing resolution. Meanwhile, as reviewer #3 mentioned, the transition could also help the adhesion of the interface (Reviewer #3, comments #3).

Regarding to this comment, we added the following Figures in SI:

Figure R3. The added Supplementary Figure S23. Optical microscope images of g-DLP printed G0 and G60 layered structure with a layer thickness of 200 μm , a) printed in the horizontal direction; b) printed in the vertical direction. The scale bar is 200 μm .

5. Comments on: Additional work

Comments #5.1: The manuscript is technically sound and requires little additional work to be publishable in a specialized journal. A few extra tasks may help the readership better appreciate its novelty.

Response: We thank the reviewer's positive feedback on this work and the constructive comments and suggestions for help improve the quality of the manuscript.

Comments #5.2: The authors' claim that their work represents a general strategy of rationally designing novel inks may be better supported by including more than one exemplary resin formulation.

Response: We thank the reviewer's suggestion. Following in the same strategy, we add an exemplary resin formulation with the generally used monomers that were also able to generate organogel under low DoC state with rich hydrogen bond moieties. It displayed similar properties as a stretchable organogel at low DoC and a stiff thermoset at how DoC (Figure R4; Figure S21). This figure was added in SI.

Figure R4. The added Supplementary Figure S21. a) Chemical structure of monomers used for exemplary resin formulation; Strain-stress curves of g-DLP printed exemplary resin of b) stiff thermoset state printed in G0; c) rubbery organogel state printed in G60.

Comments #5.3: The authors may consider naming a few applications (other than fast-prototyping) that are not susceptible to the difficulty of preserving internal property contrast over a longer term.

Response: We thank the reviewer for this comment. As we discussed in the manuscript, it can be used for validation of smart composite design and the actuators and sensors can also be practically used as disposable gadgets considering the fast printing speed. Besides, an ongoing work we are doing to use it to print shape morphing structures that require the printed structures to be foldable or stretchable before deploying and stiff after deploying (for example, the vessel stent). For such an application, it needed multi-properties at the beginning to control the deformation but not need the property contrast after deploying (the UV post-processing can turn the gel part into stiff thermoset and fix the deform).

Comments #5.4: The mechanical properties of most demonstrations are binary (stiff + elastic), with the exception of the sequential deformation designs (ternary, elastic). The authors' claim that the ink will help obtain a continuously varying modulus (Pg. 24) may be better supported by presenting an example with a smooth internal property gradient. However, this may require a better understanding of the glass transition behavior?

Response: We thank the reviewer for bringing up this good point. Following the reviewer's suggestion, we have added the demonstration with a continuous gradient sample from G0 to G70 as shown in Figure R5a. The sample was programmed at 90 °C into a helix shape and fixed by rapid cooling (Figure R5b). Afterward, the programmed sample was put into an 80°C water bath for free recovery. As the lower T_g resulting faster recovery, the sample displayed a continuously recovering from the G70 (lowest T_g) part to G0 (highest T_g) part (Figure R5c). This demonstration was added as Supplementary Movie 1.

Figure R5. The added Supplementary Figure S16. a) g-DLP printed sample with a continuous gradient; b) Sample with continuous gradient was programmed into a helix shape at 90°C and fixed by rapid cooling; c) Shape recovery of the programmed sample in 80°C water bath.

We also added the following discussion in g-DLP printing for composites structures section (highlighted on page 12 of the word file):

"To verify the capability of the single vat multi-properties g-DLP printing, we print a sample with continuous gradient grayscale from G0 to G70. The sample is expected to have a continuous Tg transition from ~90 °C to ~ -10 °C. Thus, we exploit the shape memory feature of thermoset polymer to program the sample into a helix shape at 90 °C and fix the shape by rapid cooling in ice water. Afterward, the programmed sample

is dropped into an 80 °C water bath for free recovery. The sample exhibits sequential shape recovery from the low Tg region to the high Tg region, which reflected the smooth transition of multi-properties. This experiment is presented in Supplementary Movie 1 and Supplementary Fig. S16."

Comments #5.5: Identifying the dose-DoC correlation and in particular the dose threshold for glass transition in terms of energy per unit ink volume (instead of greyscale%) will definitely help readers reproduce the results.

Response: We thank the reviewer for the comment. We have followed the reviewer's suggestion and added the following table in Support Information to help the readers reproduce the results.

Response table 1. Added Table S1. Detailed grayscale and their corresponding RGB value, light intensity and volumetric energy input.

Grayscale	RGB value	Printer light intensity mW/cm ²	Total energy per volume mJ/mm ³
G0	255	24.4	14.64
G10	229	21.1	12.66
G20	204	17.6	10.56
G30	179	14.3	8.58
G40	153	9.9	5.94
G50	128	5.5	3.3
G60	102	3.2	1.92
G70	77	1.4	0.84

Reviewer #2

General comments:

The authors describe a straight forward approach to fabricating materials that possess controllable stiffness using greyscale 3D printing. Although the work uses common monomers and crosslinkers, the combination of precursors used is unique and the formulation yields materials with tunable stiffnesses based on the light intensity (degree of polymer conversion). The authors are able to demonstrate high contrast in material stiffness, beyond what has previously been demonstrated due to an extensive network of hydrogen bonding. With this new ability to spatially control material stiffness with high contrast in 3D objects, the group demonstrate a number of applications, including airless tires, strain sensors, biomimetic arteries, heart valve, nacre, fish fin, pneumatic actuators.

The authors are careful not to oversell the work and highlight some of the challenges, such as the presence of unreacted acrylate groups that lead to the materials UV-sensitivity, layer artifacts and the limitation in applications due to print resolution. The authors offer some potential solutions to these challenges.

Response: We thank the reviewer for the positive comments about our work and for taking the time to carefully review the manuscript. In the following, we provide point-by-point response to the comments from the reviewer.

Comments #1: The formulation contains 20% w/w of urethane crosslinker. This is a very low amount of crosslinker for a 3D printable formulation. The network of hydrogen bonding presumably helps in providing structural integrity to the materials for it to be printable. At a greyscale of 60, the polymer has a polymer conversion of ~60%. Given the polymerization reaction is chain growth, and the above low crosslinker concentrations, there is a high concentration of unreacted monomer and/or short oligomers. The isobornyl acrylate has a relatively low boiling point and does not hydrogen bond and would thus be susceptible to becoming volatile. Can the authors comment on the volatility or even the loss of materials through handling the printed objects given that the materials is likely made up low molecular weight species.

Response: We would like to thank the reviewer for raising this very good point. The two small molecule precursors we used 2-hydroxyethyl acrylate and isobornyl acrylate have a boiling point of 200 °C and 120 °C respectively. As the gel state is a hydrogen bonding dominated network and the ester bond of the isobornyl acrylate is also contributing to hydrogen bond with the network, which greatly decrease the volatility of uncured isobornyl acrylate monomers. Covered container or low temperature could further help longer time preservation. In Figure S10, the G60 gel sample was stored in a drawer at room temperature for 10 days and the properties was well preserved. Meanwhile, we are exploring alternative monomers that include monomers with a higher boiling point as the stiff block.

Comments #2: Can data points corresponding to this work be included in Figure 2f?

Response: We thank the reviewer for the suggestion. We have added the data points in to Figure 2f. Also following reviewer #1's comments #3.3, we have updated the Figure 2f.

Comments #3: Can these work be included in Figure 2f• Tingting Zhao, Ran Yu, Shan Li, Xinpan Li, Ying Zhang, Xin Yang, Xiaojuan Zhao, Chen Wang, Zhichao Liu, Rui Dou, and Wei Huang ACS Applied Materials & Interfaces 2019 11 (15), 14391-14398 DOI: 10.1021/acsami.9b03156.

Response: We thank the reviewer for providing this related paper. We have included it in Figure 2f (see below Figure R6 also) accordingly.

Figure R6. The updated Figure 2f) a comparison of the mechanical property range between the single vat g-DLP printing with generally reported or commercially available DLP ink.

Comments #4: Correct the typos in lines 424- 427. These gels generally rely on hydrogen bonding between solvent molecules and polymers. Here, the g-DLP ink formulation is inspired by these gels. At low conversion, monomers are able to perform like solvent molecules in organogel via hydrogen bonding, while at high conversion, monomers crosslink into a stiff thermoset.

Response: We do appreciate the reviewer for catching this typo and giving the correction. We have revised the content accordingly and carefully checked the manuscript to ensure the quality of the writing.

Comments #5: Can the authors add a description of how the objects are processes/cleaned post-printing. Given point #1, this may influence the properties of the objects.

Response: We thank the reviewer for bringing up this valuable feedback. **All samples are raised with isopropyl alcohol after printing and wrapped in aluminum foil for further testing or demonstration.** We have added this information (below) to materials and method section on page 27.

“All samples are raised with isopropyl alcohol after printing and wrapped in aluminum foil for further testing or demonstration.”

Reviewer #3

General comments:

Qi and co-workers report on the fabrication of multi-material 3D structures using grey scale stereolithography. By careful selection of the acrylate-based monomers, they were able to vary the mechanical properties over a broad range by printing with a single vat. In contrast to their previous work, the newly developed material does not require a thermal post-curing step and benefits from a distinctive extension of the elongation. Thus, in my opinion the topic of the manuscript makes a significant contribution to the field and will be of interest for a broader scientific community as multi-material structures are relevant for numerous research fields. This was also demonstrated by the authors, who printed multi-material 3D structures applicable for soft robots/actuators or electronics

Response: We do appreciate the reviewer for giving the high opinion of our work with the statement of "the topic of the manuscript makes a significant contribution to the field and will be of interest for a broader scientific community as multi-material structures are relevant for numerous research fields." We also thank the reviewer for taking the time to carefully review our manuscript and provide the constructive comments and suggestions. Following, we provided point-by-point response.

Comments #1: The authors showed that the printed samples retain their mechanical properties over several days at room temperature in a light protected environment. Along with light, the partly cured networks are also expected to be sensitive to higher temperature. Did the authors study if higher temperatures ($> 70\text{ }^{\circ}\text{C}$) affect the multi-material properties?

Response: We thank the reviewer for bringing up this important question. The reviewer is correct. The g-DLP printed sample is also sensitive to high temperature. The organogel is essentially a semi-solid system with organic liquid phase (in this case, is the uncured monomers) immobilized in the gel network. High temperature weakens the hydrogen bonding and makes liquid phase volatile, just like any organogels or

hydrogels.

We added the following discussion in the **discussion** section (highlighted on page 27 of the word file):

"Firstly, the g-DLP printed rubbery part is an organogel with liquid phase (uncured monomers) immobilized in the hydrogen bonding dominated gel network. Just like general organogels, it is sensitive to high temperatures since it weakens the hydrogen bonding and makes the liquid phase volatile."

Comments #2: The authors comprehensively discuss the limitations of their system in the manuscript. Regarding the resolution, they mention a pixel scale transition layer of 50 – 100 μm , which limits the resolution of the developed resin. However, I am missing the actual resolution of the multi-material domains in the manuscript and I recommend that the authors address this point in their work.

Response: We appreciate the reviewer for bringing up this good point. The layer thickness is 50 μm and pixel size is also at 50 μm scale. There is a transition due to the inevitable diffusion. We added the following optical microscope images in supporting information as Figure S23. G0 and G60 laminated samples with a layer thickness of 200 μm were printed on two directions as shown in the Response Figure 2. The G0/G60 interface was more clear on the vertical direction with thinner transition less than 50 μm (1pixel) as shown in Figure R7a. However, the diffusion induced transition was more clear on the horizontal direction(Figure R7b). The transition thickness was around 50 to 100 μm (1-2 pixel). This could be optimized further with different approaches like projector with smaller pixel size, algorithm corrected grayscale images or resin with more photo absorber.

Figure R7. The added Supplementary Figure S23. Optical microscope images of g-DLP printed G0 and G60 layered structure with a layer thickness of 200 μm , a) printed in horizontal direction; b) printed in vertical direction. The scale bar is 200 μm .

We added the following content in the **discussion** section (highlighted on page 27 of the word file):

"Although such a transition layer is negligible for all the demonstrated printing size scales in this work, it could be optimized further with different approaches like using projector with smaller pixel size, algorithm-corrected grayscale images, or resin with more photo absorber."

Comments #3: Whilst the transition layer negatively affects the resolution, it might contribute to a higher adhesion strength between the soft and the hard part of the printed structures. The authors should address and comment on the interlayer adhesion between the soft and rigid domains, which is often a crucial issue in 3D printed multi-material structures.

Response: We thank the reviewer for bringing up another good point on the transition layers. We agree with the reviewer the transition layer might contribute to the adhesion strength. Since the multi-material structure is generated from the same resin, the adhesion is expected to be good. We have added the following experiment in Supporting Information, and address the interlayer adhesion between soft and rigid domains in the revision as the reviewer suggested. As shown in Figure R8 (Supplementary Figure S17), the energy need to break the interface between the soft (G60) and stiff (G0) domains is greater than the energy needed to break the soft part itself.

Figure R8. The added Supplementary Figure S17. Stretching and breaking of g-DLP printed G0 and G60 composite.

We also added the following content in the g-DLP printing for composites structures section (highlighted on page 12-13 of the word file):

"Moreover, we further examine the interface and adhesion between the stiff thermoset and rubbery organogel. As the different curing states are all generated from the same resin, the adhesion is expected to be good. Meanwhile, the inherent transition between different curing states due to diffusion enhances the adhesion between different domains of the g-DLP printed structures. As presented in Supplementary Figure S17, the energy needs to break the interface between the soft (G60) and stiff (G0) domains is greater than the energy needs to break the soft part itself, demonstrating the good adhesion."

Comments #4: In the experimental part, the reference resins are missing.

Response: We thank the reviewer for catching this missing information. We have updated the detailed reference resins information in SI supplementary text section accordingly.

"Section 3. Reference resins formulation used in the work

1. Figure S11, control resin without 2-HEA: Isobornyl acrylate (Sigma-Aldrich, MO, USA) and AUD (Ebecryl 8413, Allnex, GA, USA) are mixed with the weight ratio of 3:1. Then 1wt% photoinitiator (Irgacure 819, Sigma-Aldrich, MO, USA) and 0.05wt% photo absorber (Sudan I, Sigma-Aldrich, MO, USA) are added and mixed well.
2. Figure S13, control resin with poor hydrogen bond moieties (Figure Sx): Poly(ethylene glycol) diacrylate (PEGDA, Sigma-Aldrich, MO, USA), butyl acrylate (BA, Sigma-Aldrich, MO, USA) and isobornyl acrylate (Sigma-Aldrich, MO, USA) are mixed with the weight ratio of 1:3:1. Then 1wt% photoinitiator (Irgacure 819, Sigma-Aldrich, MO, USA) and 0.05wt% photo absorber (Sudan I, Sigma-Aldrich, MO, USA) are added and mixed well.
3. Figure S21, alternative resin with rich hydrogen bond moieties: 2-[[[(Butylamino)carbonyl]oxy]ethyl acrylate (Sigma-Aldrich, MO, USA), acrylate acid (Sigma-Aldrich, MO, USA) and AUD (Ebecryl 8413, Allnex, GA, USA) are mixed with the weight ratio of 2:2:1. Then 1wt% photoinitiator (Irgacure 819, Sigma-Aldrich) and 0.05wt% photo absorber (Sudan I, Sigma Aldrich) are added and mixed well."

Reviewers' Comments:

Reviewer #1:

Remarks to the Author:

The revision has satisfactorily addressed concerns raised in my review report. In particular, I appreciate that the authors elaborated on the uniqueness of their photo-ink when it comes to printing complex functional constructs. In this regard, I believe making the digital designs of these constructs available to the readership will be invaluable in ensuring the novel work's reproducibility. Please point the readers to where the digital forms (e.g., in *.STL or *.OBJ) of the tested structures are deposited, or submit them as SI.

I do not have any other comment. Congrats on a great piece of work.

Reviewer #2:

Remarks to the Author:

The authors adequately responded to my concerns.

Reviewer #3:

Remarks to the Author:

The authors addressed all questions and suggested revisions satisfactorily.

Responses to Reviewers

Reviewer #1

The revision has satisfactorily addressed concerns raised in my review report. In particular, I appreciate that the authors elaborated on the uniqueness of their photo-ink when it comes to printing complex functional constructs. In this regard, I believe making the digital designs of these constructs available to the readership will be invaluable in ensuring the novel work's reproducibility. Please point the readers to where the digital forms (e.g., in *.STL or *.OBJ) of the tested structures are deposited, or submit them as SI.

I do not have any other comment. Congrats on a great piece of work.

Response: Those structures were printed with the images generated directly by MATLAB (Fig.3c-d, 4c) or SOLIDWORKS (Fig.1f, 3b,4a-b,5a-g,6a-c) with MATLAB processing to add the grayscale features. We will add those files on our group website: <https://www.msm.gatech.edu/> once published. We also welcome readers to contact the corresponding author for the designs. We added the following statement in the Data Availability section:

“All other data underlying the results including those digital designs of those printed structures of this study are available from the corresponding author upon request.”

Reviewer #2:

The authors adequately responded to my concerns.

Reviewer #3:

The authors addressed all questions and suggested revisions satisfactorily.

Response: We would like to thank all the reviewers again for talking the time to carefully review the paper and helping us improve the quality of the manuscript.